

# Instrument observation strategy of new generation three-axis stabilized geostationary meteorological satellite of China

Jian Shang[1], Pan Huang[2], Huizhi Yang[2], Chengbao Liu[1], Jing Wang[1], Lei Zhao[1], Shengxiong Zhou[3], Xiaodong Chen[2], Lei Yang[1], Zhiqing Zhang[1]

[1] National Satellite Meteorological Center, China Meteorological Administration, Beijing, 100081, China
[2] Beijing Institute of Electrical and Mechanical Engineering, Beijing, 100074, China
[3] Southwest Electronics and Telecommunication Technology Research Institute, Chengdu, 610000, China

*Correspondence to*: Lei Yang (yangl@cma.gov.cn), Zhiqing Zhang (zqzhang@cma.gov.cn)

**Abstract.** Fengyun-4 (FY-4) satellite series is the new generation of geostationary meteorological satellite of China. Thenewly adopted three-axis stabilized attitude control platform can increase observation efficiency and flexibility, while bringing great challenge to image navigation as well as integrated observation mode design. Considering the requirements of the earth observation, navigation and calibration besides observation flexibility, instrument observation strategies are proposed, including the earth, the moon, stars, cold space, blackbody, diffuser observations, on which the instruments' in-orbit daily observations must be based. The most complicated part is star observation strategy, while navigation precision is dependent on in-orbit star observations. Flexible, effective, stable and automatic star observation strategy directly influences obtaining star data and navigation precision. According to the requirement of navigation, two specific star observation strategies for the two main instruments onboard FY-4 were proposed to be used in the operational ground system. The strategies have been successfully used in FY-4 in-orbit test for more than a year. Both the simulation results and in-orbit application results are given, including instrument observation strategies, star observation strategies and moon tasks, to demonstrate the validity of the proposed observation strategies, which lay important foundations for the instruments' daily operation.

## 1 Introduction

Fengyun (FY) series meteorological satellite, operated by National Satellite Meteorological Center, China Meteorological Administration, has been playing an important role in meteorological forecast, cloud detection, precipitation measurement, and so on. Fengyun-4 (FY-4) is China's new generation geostationary meteorological satellite series, which is three-axis stabilized instead of spin stabilized as in Fengyun-2 (FY-2) satellites. The three-axis stabilized attitude control mode can absolutely increase observation efficiency and flexibility (NOAA and NASA, 2005). However, it brings great challenge to image navigation and registration (INR) compared with spin-stabilized satellite, which can satisfy the navigation requirement just by using edge detection of the earth's disk (Lu et al., 2008). Image navigation is an essential and fundamental component in the data processing of geostationary meteorological satellites. The purpose of image navigation is

to acquire each image pixel's latitude, longitude, height, and other important parameters. The uncertainty in the instrument's line of sight is the main error source for navigation (Yang and Shang, 2011). The space thermal source changes enormously and the main forms of thermal conduction are radiation and conduction. Lacking of important convection makes the thermal environment of space orbit very abominable. A spin stabilized satellite tends to equalize the thermal variation seen by the

instrument over the day, whereas the thermal gradients across the three-axis stabilized platform are more extreme (Li et al., 2007; Harris and Just, 2010). The INR challenge of three-axis stabilized geostationary satellite comes from thermal elastic deformation from the solar source, launch violation, orbit measurement error, attitude measurement error, and so on. The misalignment caused by thermal elastic deformation, which cannot be measured directly, is the most difficult element to model (Shang et al., 2017). The misalignment must be calculated and forecasted accurately in the ground system. With

uploaded coefficients, compensation could be accomplished by the on-board system. Thus navigation is fulfilled by complicated satellite-earth integrated operation (Li and Dong, 2008; DOC et al., 2014a; DOC, 2014b).

Star sensing using the instruments onboard the satellite sheds new light on image navigation of three-axis stabilized geostationary satellites. It has many advantages versus landmark navigation, which has been used widely in remote sensing image processing. Stars are ideal point sources and their position on the celestial sphere can be determined precisely. Stars

can be observed in both day and night, and can obtain the line-of-sight directly (Li et al., 2007). Before star navigation, there is much work to be carried out, among which establishing star observation strategies for different instruments is fundamental and difficult. The strategies, which are constrained by a lot of conditions, need to select the most proper stars from a number of stars observed in the instrument's field of view, within the given observation time determined by instrument time schedule. Then based on star observation strategies, considering the integrated requirements of navigation, calibration as well as the

earth observation, instrument observation strategies should be developed to guide the instruments to automatically carry out every in-orbit observations.

This paper is a foundation of FY-4 image navigation work concerning task planning. It focuses on the instrument observation strategies as well as flexible and automatic star observation strategies establishment of FY-4A main instruments. Simulations are carried out to give long time analysis. The first satellite of FY-4 satellite series, FY-4A, has been launched

on December 11, 2016. In-orbit test results are obtained and used to demonstrate the strategies' effectiveness, which pave the way of high accuracy navigation of FY-4A.

**2 Instrument Observation Strategies**

Instruments' in-orbit observationsare more flexible than FY-2 satellite, as FY-4 uses three-axis stabilized platform instead of spin-stabilized platform, and thus more complicated. Towards the two main instruments aboard FY-4, the Advanced

Geosynchronous Radiation Imager (AGRI) and Geosynchronous Interferometric Infrared Sounder (GIIRS), instrument observation strategies are designed to figure out their in-orbit observation task modes, making full use of the observation time and flexibility.



## 2.1 AGRI Observation Strategy

AGRI's main object is to carry out high temporal and spatial resolution imaging of the full disk. Besides, higher temporal resolution imaging of China region and certain interesting regions is very important for regional weather forecast and disaster monitoring. And hemisphere observation should also be arranged as required. From navigation and registration's perspective, periodical star observation, landmark observation and moon observation are obligatory. From calibration's perspective, blackbody, cold space, diffuser and moon must be observed. In all, 17 observation modes are designed for AGRI as listed in Table 1, considering high sensitivity scan as well as normal scan with regard to the earth observation (Shang et al., accepted). Nowadays full disk observation and China region observation are chosen as the main earth observation modes. In every 15 minutes, one full disk task or three China region tasks, one blackbody task and one star task are arranged in the time schedule, which is used by the whole ground system to operate on schedule. Star observation strategy is the most complicated part of instrument observation strategy, which will be proposed below in detail. Landmark and cold space observation are contained in full disk task automatically. This guarantees routine earth observation along with fundamental requirements of navigation and calibration. Diffuser task will be inserted into the time schedule while the computed solar declination angle is suitable.

**Table 1: Observation modes of AGRI.**

| No. | Observation mode | Observation type |
|-----|-----------------|------------------|
| 1 | Full disk | Normal scan / High sensitivity scan |
| 2 | North hemisphere | Normal scan / High sensitivity scan |
| 3 | South hemisphere | Normal scan / High sensitivity scan |
| 4 | China | Normal scan / High sensitivity scan |
| 5 | Region | Normal scan / High sensitivity scan |
| 6 | Moon | Normal scan / High sensitivity scan |
| 7 | Landmark | Normal scan / High sensitivity scan |
| 8 | Star | Dwell |
| 9 | Blackbody | Dwell |
| 10 | Diffuser | Dwell |

The positions of the sun and the moon in the international celestial reference system at any time are obtained from high-precision DE412 (Development Ephemeris 412) released by NASA JPL (National Aeronautics and Space Administration, Jet Propulsion Laboratory), recommended by IAU (International Astronomical Union). The positions are transformed into geocentric celestial reference system and then inertial coordinate system. If the sun is forecasted to appear in the vicinity of AGRI's field of view, its influence on star sensing, cold space observation, the earth observation as well as the motion track of scanning mirrors must be considered, in order to ensure the observation validity besides the safety of the instrument. Moon task will be inserted automatically into the time schedule while the moon is forecasted to appear in AGRI's field of view. The instrument's pointing angels of the moon are computed finally in the instrument coordinate system, which will be written into the observation instruction parameter file and uploaded to the satellite.



## 2.2 GIIRS Observation Strategy

GIIRS is another important instrument onboard FY-4A. Its main objective is to detect atmospheric temperature, moisture and trace gas content precisely, providing input data for numerical weather forecast, disastrous weather monitoring and atmospheric chemical composition detection. Therefore, GIIRS's main detection mode is dwell detection of the atmosphere.

From navigation's perspective, periodical star observation and landmark observation are obligatory. A unique ground-based laser system is proposed to assist GIIRS navigation. From calibration's perspective, blackbody, cold space and the moon must be observed. In all, 7 observation modes are designed for GIIRS as listed in Table 2. In every 15 minutes, one region task, one blackbody task, one cold space task and one star task are arranged in the time schedule, which guarantees routine earth observation along with fundamental requirements of navigation and calibration. Landmark task is arranged periodically

to estimate navigation precision, interrupting routine region task, and thus can't be carried out frequently. In addition, moon task will be automatically inserted while the moon is forecasted to appear in GIIRS's field of view, which is a little different from AGRI and should be forecasted respectively. Also the sun must be forecasted in advance to ensure the observation validity besides the safety of the instrument.

**Table 2: Observation modes of GIIRS.**

| No. | Observation mode | Observation type |
|-----|------------------|------------------|
| 1 | Region | Step-dwell |
| 2 | Moon | Dwell |
| 3 | Landmark | Step-dwell |
| 4 | Star | Dwell |
| 5 | Laser | Step-dwell |
| 6 | Blackbody | Dwell |
| 7 | Cold space | Dwell |

## 3 Star Observation Strategies

The two main instruments aboard FY-4A are both designed with the ability to sense stars of magnitude at least lower than 6.0, to help to achieve high accuracy of image navigation in three-axis stabilized attitude control platform. The ground system first forecasts all the stars that will appear in the instrument's field of view at the given time. Then proper stars are

chosen according to complicated star observation strategy exclusively developed for the instrument. The star observation instructions are then automatically generated and uploaded to the satellite. For star sensing, the instrument is commanded to dwell at angles determined by the ground system for a given star crossing. The instrument inertial viewing angles then drift at earth rotation rate in a roughly west direction as the spacecraft attitude control continuously aligns the instrument boresight with the earth (Gibbs et al., 2008).

### 3.1 AGRI StarObservation Strategy

AGRI is the core instrument aboard FY-4A satellite, which aims to carry out high temporal and spatial resolution imaging in 14 spectral bands in visible (VIS), near infrared (NIR) and infrared (IR) spectral regions (Zhang et al., 2016; Dong, 2016). The one-pixel-accuracy navigation of AGRI is one of the most difficult tasks in satellite-earth integrated operation. Thus,

star navigation is indispensable. AGRI has 14 observation bands, the second (0.55-0.75μm) of which is designed to sense stars of magnitude higher than 6.0. The detector size is 32 (north-south direction) by 4 (west-east direction), with a gap between adjacent columns, which must be considered in developing AGRI's star observation strategy to guide its in-orbit daily star sensing.

Star observation strategy of AGRI includes several aspects. Firstly, the frequency of star sensing must be determined, which

is mainly determined by the changing regularity of thermal elastic deformation of the satellite platform as well as the instrument, while balancing among different observation tasks. Secondly, the requirements of star centroid extraction must be considered. Thirdly, the requirements to ensure the accuracy of thermal elastic deformation calculation, which is the key issue in image navigation of three-axis stabilized geostationary satellites, must be considered. The criteria of choosing optimal stars for AGRI are proposed as follows.

(1) The main task of AGRI is to carry out the earth observation. Star sensing as an indispensable part of image navigation, should have the priority to be arranged periodically besides the earth observation tasks.

(2) The gap between adjacent columns is a big disadvantage to star centroid extraction. Thus dwell observation mode is recommended for star sensing, waiting for the star crossing the whole focal plane, aiming at obtaining star observation data of relative long time series. This can effectively improve the accuracy of star centroid extraction (Zhang et al.,

2018).

(3) Considering AGRI's angular resolution, the candidate stars should be apart from each other.

(4) The star with lower magnitude should be considered in priority, which is advantageous for star centroid extraction and thus navigation.

(5) The magnitude of target star must be lower than AGRI's observation ability. Meanwhile, it should be higher than the

25 magnitude threshold that may make the image saturated, which will affect normal star centroid extraction.

(6) The target star should always be within AGRI's field of view, both at the start and at the end of the observation.

(7) The target star should not be shaded by the earth atmosphere, the sun, the moon, etc.

(8) During the observation, the target star must not be shaded by the earth itself.

(9) The observation of the target star should not be affected by the stars that newly entered into the field of view or newly

traced out of the earth.

(10) Exclude variable star that may have disadvantageous effect on star centroid extraction.

(11) Double stars should be selected carefully, balancing the number of proper stars against the effect on star centroid extraction.



(12) Finally, and the most importantly, the distribution of candidate stars should be strictly constrained to ensure the accuracy of thermal elastic deformation parameter calculation.

The known parameters include AGRI's field of view, angular resolution, the detector size, the given time and time limit of star sensing, and so on. A variety of thresholds needs to be calculated using these parameters, including minimum resolution threshold, minimum magnitude threshold, maximum magnitude threshold, sun/moon/earth effect threshold, moving threshold, variable star threshold, double star threshold, distribution threshold, etc.

The most important distribution threshold should be determined on the basis of angle distance of two stars. The method to compute angle distance from instrument pointing angles is derived as follows. A plane coordinate system is established in Fig. 1. The coordinate origin is located in the center of the earth. The positive directions of the X axis and Y axis are west and north, respectively. Points $S$ and $O$ denote the satellite and the earth center, respectively. $a$ and $b$ denote two stars, whose viewing vectors intersect with plane $XOY$ at point $A$ and $B$. $A_X$, $A_Y$, $B_X$, and $B_Y$ are the projection points of the vectors on X and Y axis, respectively. The scan angles in east-west direction and step angles in north-south direction of observing star a and b are $\alpha_A$, $\beta_A$, $\alpha_B$ and $\beta_B$.

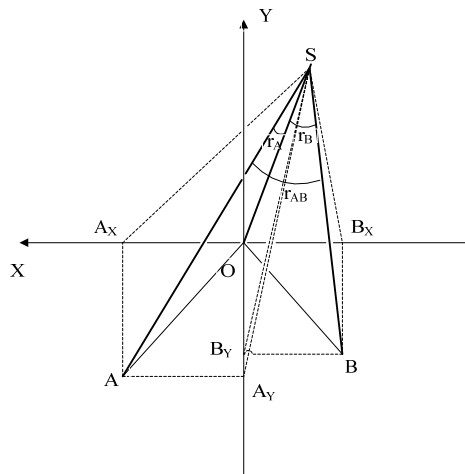

**Figure 1: Coordinate system established in computing angle distance.**

Angle $\gamma_A$ and $\gamma_B$ of star a and b are computed as in Eq. (1) and (2):

$$\tan \gamma_A = \frac{OA}{OS} = \frac{\sqrt{OA_X{}^2 + OA_Y{}^2}}{OS} \tag{1}$$

$$\tan \gamma_B = \frac{OB}{OS} = \frac{\sqrt{OB_X{}^2 + OB_Y{}^2}}{OS} \tag{2}$$

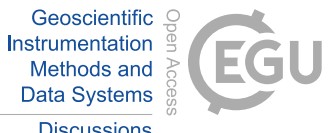



where $OS$ is the distance from the satellite to the earth center. From Eq. (3) and (4), angle $\gamma_A$ and $\gamma_B$ can be obtained using Eq. (5) and (6):

$$tan\,\alpha_A = \frac{OA_X}{OS}, \quad tan\,\beta_A = \frac{OA_Y}{OS} \tag{3}$$

$$tan\,\alpha_B = \frac{OB_X}{OS}, \quad tan\,\beta_B = \frac{OB_Y}{OS} \tag{4}$$

$$tan\,\gamma_A = \sqrt{(tan\,\alpha_A)^2 + (tan\,\beta_A)^2} \tag{5}$$

$$tan\,\gamma_B = \sqrt{(tan\,\alpha_B)^2 + (tan\,\beta_B)^2} \tag{6}$$

Then the angle distance $\gamma_{AB}$ of star a and b is computed using cosine theorem, as in Eq. (7):

$$cos\,\gamma_{AB} = \frac{SA^2 + SB^2 - AB^2}{2 \cdot SA \cdot SB} \tag{7}$$

where

$$SA = \frac{OS}{cos\,\gamma_A}, SB = \frac{OS}{cos\,\gamma_B}$$

$$AB^2 = OA^2 + OB^2 - 2 \cdot OA \cdot OB \cdot cos\,\angle AOB$$

$$OA = \sqrt{OA_X{}^2 + OA_Y{}^2}, \quad OB = \sqrt{OB_X{}^2 + OB_Y{}^2}$$

$$cos\,\angle AOB = \frac{\overrightarrow{OA}\,\overrightarrow{OB}}{|OA| \cdot |OB|} = \frac{OA_X \cdot OB_X + OA_Y \cdot OB_Y}{OA \cdot OB} \tag{8}$$

### 3.2 GIIRS Star Observation Strategy

Another important instrument aboard FY-4A is GIIRS, which is the first one flying in geostationary orbit and will provide high spectral resolution IR sounding observations over China and adjacent regions (Zhang et al., 2016). The accuracy of GIIRS image navigation is also one pixel, and star navigation is introduced into GIIRS navigation flow. GIIRS is designed deliberately with a visible band (0.55-0.75μm) to sense stars of magnitude lower than 6.5. Star observation strategy of GIIRS must also be developed, considering both the similarities and differences between AGRI and GIIRS. The similarities include requirements of star centroid extraction as well as thermal elastic deformation parameter calculation. The main difference comes from the different instantaneous field of view (IFOV), which will be analyzed in detail.

(1) AGRI's IFOV is 448 *μrad* (north-south direction) by 98 *μrad* (west-east direction), while GIIRS's IFOV is 18480 *μrad* (north-south direction) by 14336 *μrad* (west-east direction). Thus, most of the time AGRI can only observe one star, while GIIRS has the ability to observe two or three stars, sometimes even more. The number of stars which can be observed by AGRI and GIIRS in the 21 degree (north-south direction) by 23 degree (west-east direction) field of view are obtained through simulations, and the statistical results are given in Table 3 and 4, respectively. The maximum magnitude analyzed is 7.0. Only a few groups of double stars are found for AGRI, while the percentage of multi stars for GIIRS is up to 16%. This is advantageous for star recognition. So multi star observation is considered with higher priority for GIIRS.

(2) In the process of choosing the optimal constellation, many aspects should be considered, including the number of stars

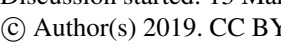



in the IFOV, the relative distribution, the minimum distance, the difference of star magnitude, etc. The first one and the second one influence the accuracy of star recognition. The third one and the forth one influence the accuracy of star centroid extraction.

(3) GIIRS detector is so large that the time consumed by the star crossing the whole detector is too long to bear. A small area of less than 10 pixels near the center of the detector is chosen to be used for star sensing, in order to avoid the effect of distortion at the edge of the detector.

**Table 3: Statistics of double stars for AGRI.**

| No. | Right ascension of star 1 / degree | Declination of star 1 / degree | Right ascension of star 2 / degree | Declination of star 2 / degree |
|-----|------------|------------|------------|------------|
| 1 | 266.148 | 2.579 | 266.142 | 2.579 |
| 2 | 184.538 | -3.954 | 184.540 | -3.949 |
| 3 | 75.136 | 3.615 | 75.141 | 3.616 |
| 4 | 234.667 | -8.794 | 234.667 | -8.791 |
| 5 | 70.896 | -8.796 | 70.894 | -8.794 |
| 6 | 97.206 | -7.034 | 97.204 | -7.033 |
| 7 | 337.207 | -0.020 | 337.209 | -0.020 |
| 8 | 133.873 | -7.971 | 133.873 | -7.970 |

**Table 4: Statistics of multi stars for GIIRS.**

| No. of stars in IFOV | Occurrence number |
|---------|---------|
| 9 | 8 |
| 8 | 18 |
| 7 | 55 |
| 6 | 43 |
| 5 | 74 |
| 4 | 398 |
| 3 | 3421 |
| 2 | 25844 |
| 1 | 160302 |
| 0 | 569397 |

The known parameters include GIIRS's field of view, angular resolution, the detector size, the given time and time limit of star sensing, and so on. A variety of thresholds needs to be calculated using these parameters, including the thresholds mentioned in AGRI's star observation strategy, as well as the threshold of star number in IFOV, the relative distribution constrain, the minimum distance and the threshold of star magnitude difference within the constellation.

## 4 Simulation Results

Before FY-4A was launched, in-depth simulations have been carried out to confirm the validity of AGRI and GIIRS star observation strategies, which are the core of instrument observation strategies. An important constrain of star observation strategy is star magnitude. The smaller the magnitude is, the higher the accuracy of star centroid extraction may achieve. The percentages of different magnitude ranges of AGRI and GIIRS are counted for every month, which are given in Table 5 and

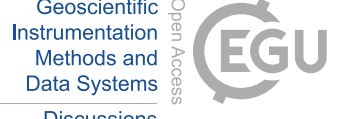



Table 6, respectively. Within the range of 3.5 to 6.5/7.5, which is determined by AGRI/GIIRS detecting ability, a few stars' magnitude lies in the range of 3.5 to 4.5, which are more likely to be selected in star observation strategy. Moreover, distribution and many thresholds mentioned above must be considered comprehensively.

**Table 5: Star magnitude statistics of AGRI.**

| Month | 3.5-4.5 | 4.5-5.5 | 5.5-6.5 |
|---|---|---|---|
| January | 9.67% | 23.37% | 66.96% |
| February | 9.75% | 23.65% | 66.61% |
| March | 9.64% | 23.47% | 66.88% |
| April | 9.63% | 23.42% | 66.96% |
| May | 9.67% | 23.52% | 66.81% |
| June | 9.77% | 23.73% | 66.50% |
| July | 9.63% | 23.43% | 66.94% |
| August | 9.77% | 23.72% | 66.51% |
| September | 9.60% | 23.47% | 66.92% |
| October | 9.70% | 23.41% | 66.90% |
| November | 9.66% | 23.58% | 66.76% |
| December | 9.80% | 23.47% | 66.73% |

**Table 6: Star magnitude statistics of GIIRS.**

| Month | 3.5-4.5 | 4.5-5.5 | 5.5-6.5 | 6.5-7.5 |
|---|---|---|---|---|
| January | 3.56% | 8.07% | 22.73% | 65.64% |
| February | 3.53% | 8.05% | 22.27% | 66.15% |
| March | 3.58% | 7.93% | 22.56% | 65.92% |
| April | 3.46% | 7.97% | 22.53% | 66.04% |
| May | 3.69% | 8.00% | 22.31% | 66.00% |
| June | 3.52% | 8.06% | 22.33% | 66.10% |
| July | 3.50% | 8.14% | 22.53% | 65.82% |
| August | 3.49% | 8.09% | 22.47% | 65.95% |
| September | 3.64% | 8.13% | 22.43% | 65.80% |
| October | 3.47% | 7.98% | 22.56% | 65.99% |
| November | 3.61% | 8.21% | 22.51% | 65.68% |
| December | 3.55% | 7.89% | 22.35% | 66.20% |

The stars of the whole year of 2017 that could be observed by the instruments are forecasted. Then optimal stars are selected for AGRI and GIIRS automatically, according to the proposed star observation strategies. In the simulation, the observation

10   time interval was set to 15 minutes. Thus, both AGRI and GIIRS should carry out 35040 star observation tasks in 2017 in all. Table 7 and Table 8 respectively present the statistical number of selected stars of AGRI and GIIRS in every month, for the convenience of display. As FY-4A flies in the geostationary orbit, the daily star observation situation is similar. For AGRI, the number of selected observable stars for each star sensing lies within 5 to 39. For GIIRS, the number lies within 7 to 51. Therefore, it can be concluded that all the cases can ensure adequate observation data needed for thermal elastic deformation

15   parameter calculation.

**Table 7: The number of AGRI selected stars in every month.**

| Month | Maximum | Minimum | Mean |
|---|---|---|---|
| January | 39 | 6 | 17.38 |





| February | 38 | 5 | 17.43 |
| March | 38 | 5 | 17.37 |
| April | 39 | 6 | 17.4 |
| May | 38 | 5 | 17.34 |
| June | 38 | 5 | 17.39 |
| July | 39 | 6 | 17.36 |
| August | 38 | 5 | 17.37 |
| September | 39 | 5 | 17.37 |
| October | 39 | 6 | 17.39 |
| November | 38 | 5 | 17.37 |
| December | 38 | 5 | 17.38 |

**Table 8: The number of GIIRS selected stars in every month.**

| Month | Maximum | Minimum | Mean |
| --- | --- | --- | --- |
| January | 50 | 7 | 20.99 |
| February | 48 | 8 | 20.96 |
| March | 51 | 8 | 20.98 |
| April | 48 | 8 | 20.94 |
| May | 49 | 8 | 20.99 |
| June | 50 | 8 | 20.96 |
| July | 50 | 7 | 21.00 |
| August | 48 | 8 | 20.96 |
| September | 51 | 8 | 21.00 |
| October | 48 | 8 | 20.97 |
| November | 49 | 8 | 20.99 |
| December | 50 | 8 | 20.97 |

A specialized software was developed to demonstrate the observable stars, in which all the observable stars as well as finally

5 selected optimal stars can be seen clearly. Fig. 2 and Fig. 3 give three different examples of AGRI and GIIRS star selection, respectively, in which five stars are set as the goal. Fig. 2(a) and Fig. 3(a) show the case of a lot of stars that can be selected. Fig. 2(b) and Fig. 3(b) show the case of neither too many nor too few. Fig. 2(c) and Fig. 3(c) show the case of only a few stars. In different cases, five stars are all selected successfully, using the proposed star observation strategies. And the relative distributions, which are one of the most important aspects in thermal elastic deformation parameter calculation, are

10 very reasonable.



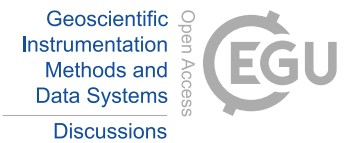

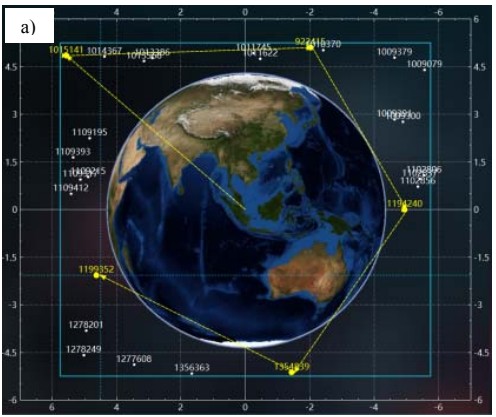

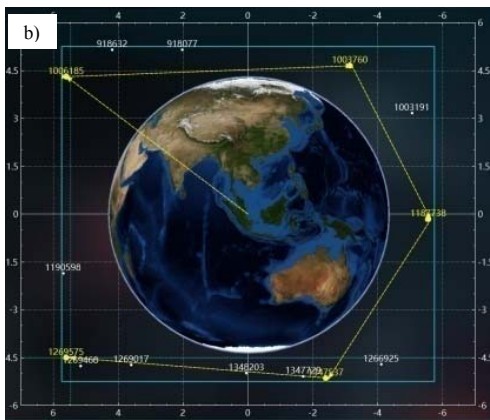





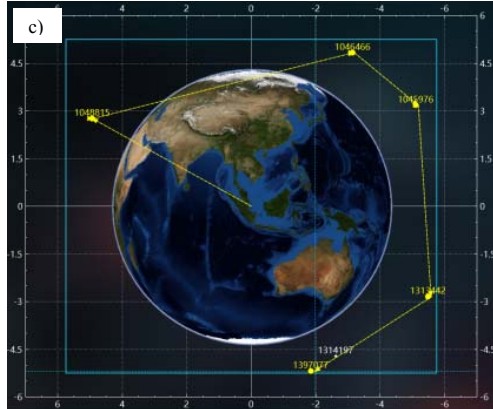

**Figure 2: Examples of AGRI star selection: (a) a lot of stars, (b) reasonable number of stars and (c) a few of stars.**

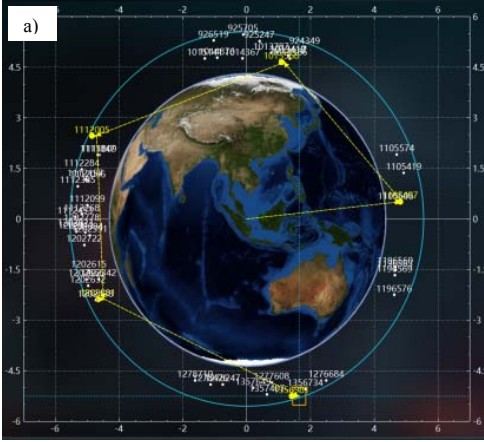





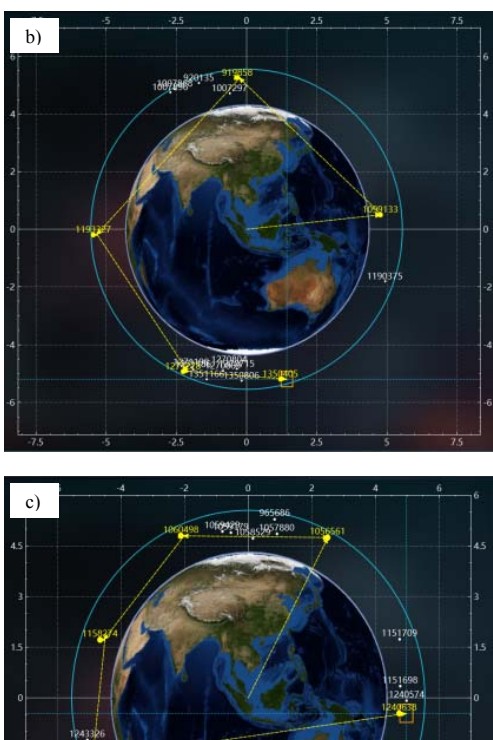

**Figure 3: Examples of GIIRS star selection: (a) a lot of stars, (b) reasonable number of stars and (c) a few of stars.**

## 5 In-orbit Application Results

Since FY-4A was launched at the end of 2016, the specifically developed instrument observation strategy software has been used in the ground system. On receiving AGRI and GIIRS's time schedules, the specially developed software firstly forecasts observable stars and selects optimal ones automatically. Then accurate execution time is computed for each star observation task, according to the laws of mirror movement, and star observation instruction parameters are generated. Other tasks,

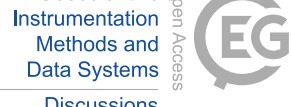



including full disk, region, blackbody,etc., are separated into the smallest pieces and accurate execution time is computed. Sun avoidance must be carefully considered while designing the mirrors' movement to protect the instruments. Finally, the complete observation instruction parameters corresponding to the whole time schedule are packed together and uploaded to the satellite. AGRI and GIIRS will carry out in-orbit observations automatically at fixed time according to the instructions.

5   The whole process, including star selection, instruction parameter generation, instruction uploading and in-orbit observation, is totally automatic and there is no need for manual intervention under normal circumstances.

From March, 2017 to March, 2018, the software of instrument observation strategy has been operating well and generated more than 163000 tasks and 1163000 instructions for AGRI, while more than 169000 tasks and 1047000 instructions for GIIRS. Fig. 4 and Fig. 5 give typical examples of AGRI and GIIRS tasks. Plenty of the earth surface, atmosphere and cloud

10  observation data have been provided to users of weather forecast, climate change, disaster monitoring, environment surveillance, and so on.

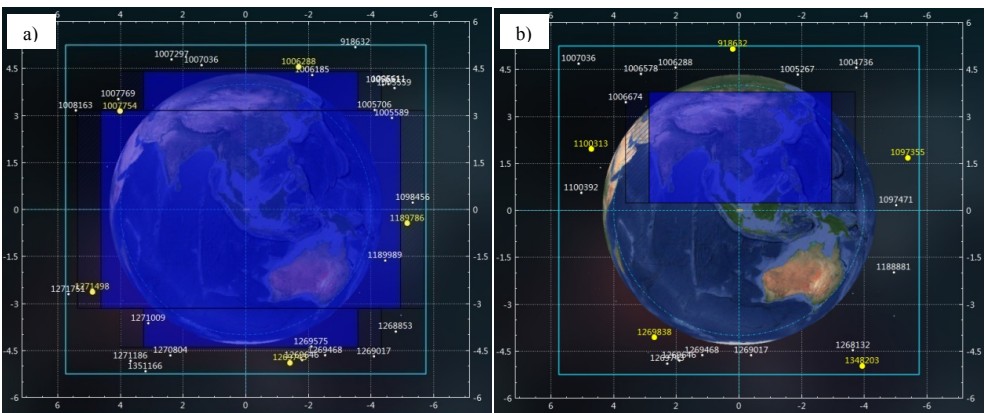

Fig. 4 AGRI typical tasks: (a) full disk observation and (b) China region observation.



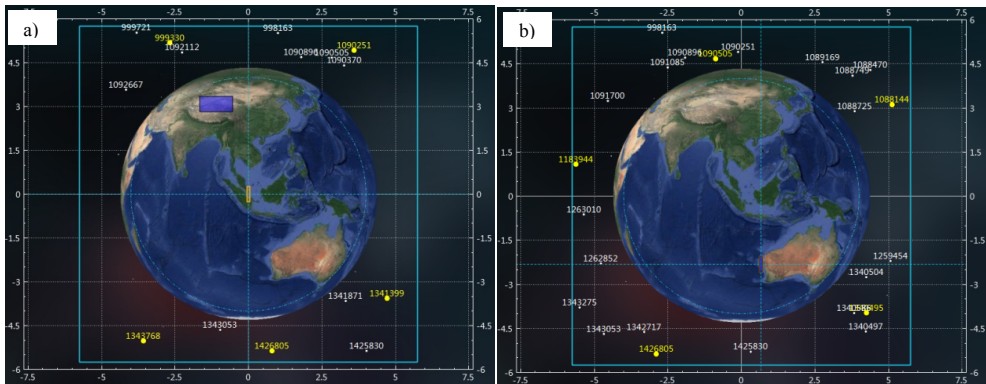

**Fig. 5 GIIRS typical tasks: (a) region observation and (b) landmark observation.**

Actual star selection results of different number of stars in the operational ground system are given as follows. Fig. 6

5 demonstrates the case that the sun is not in AGRI's field of view, when solar stray light's effect doesn't need to be considered and the useful field to select stars is relatively large. Fig. 7 demonstrates the case that the sun appears in or near AGRI's field of view, when solar stray light's effect maybe obvious and must be considered. Thus the useful field to select stars can be much smaller, resulting star selection more difficult. In this case, the requirement of a large number of stars might cannot be satisfied. Fig. 8 and Fig. 9 are different cases of GIIRS.

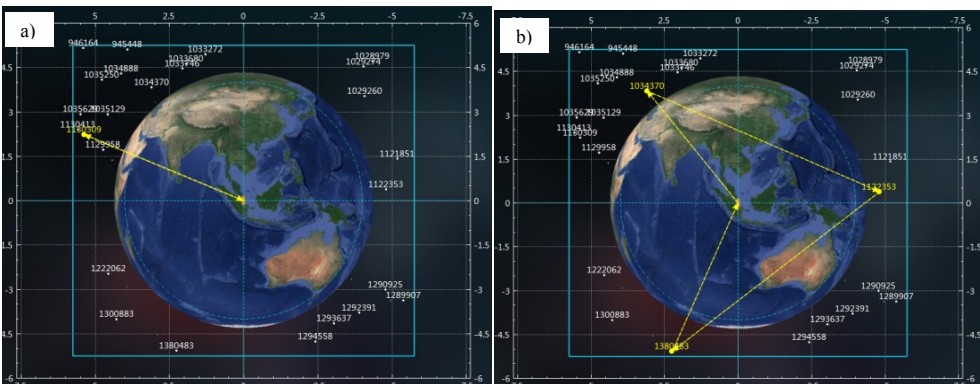





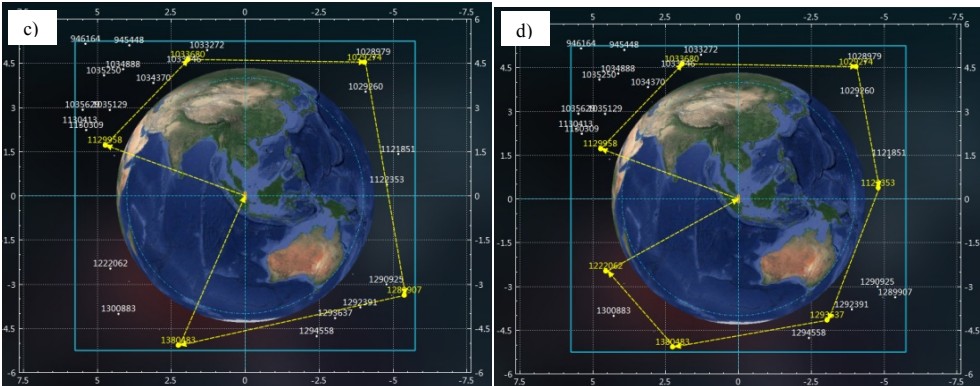

**Fig. 6 AGRI star selection results - without the sun: (a) 1 star, (b) 3 stars, (c) 5 stars and (d) 7 stars.**

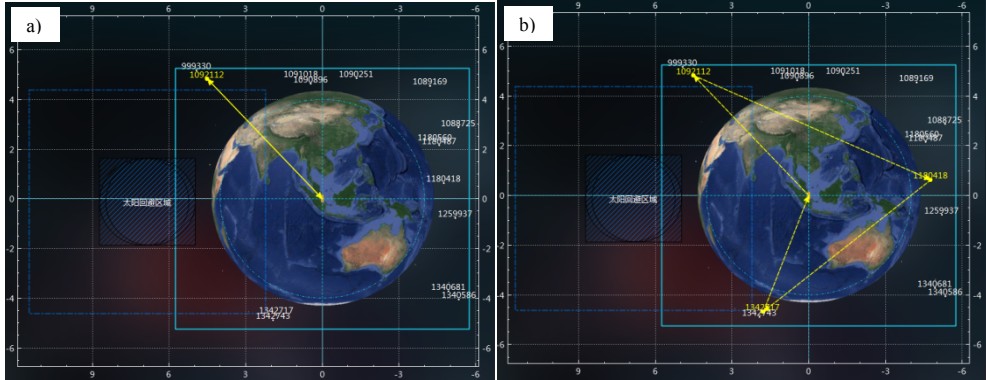





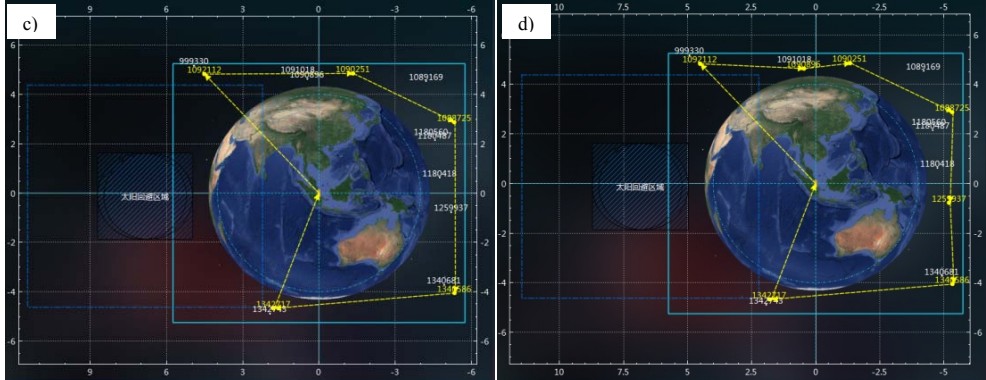

**Fig. 7 AGRI star selection results - with the sun: (a) 1 star, (b) 3 stars, (c) 5 stars and (d) 7 stars.**

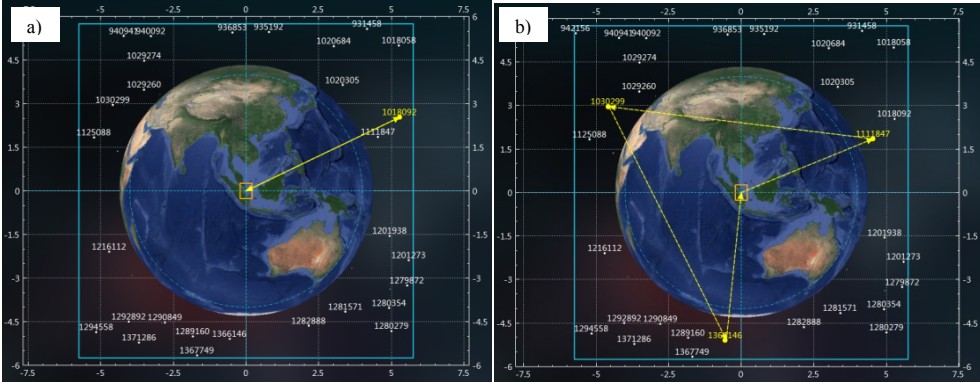





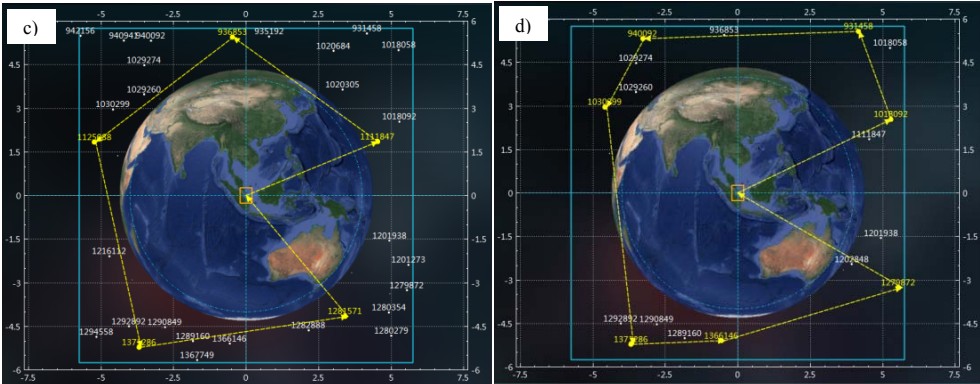

**Fig. 8 GIIRS star selection results - without the sun: (a) 1 star, (b) 3 stars, (c) 5 stars and (d) 7 stars.**

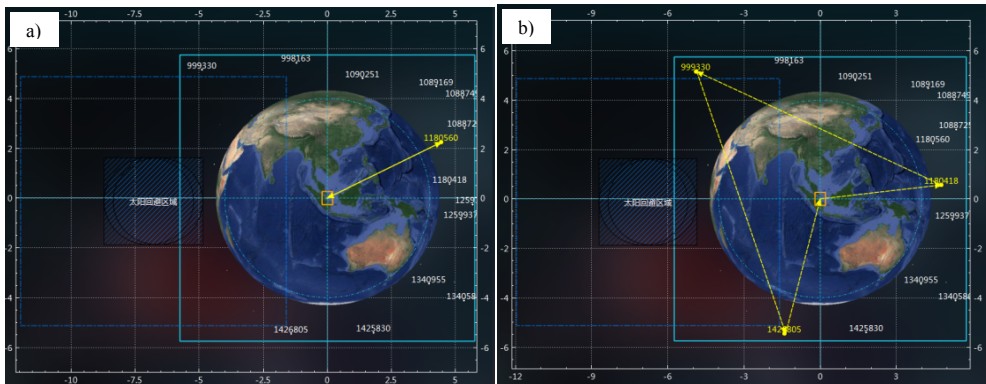





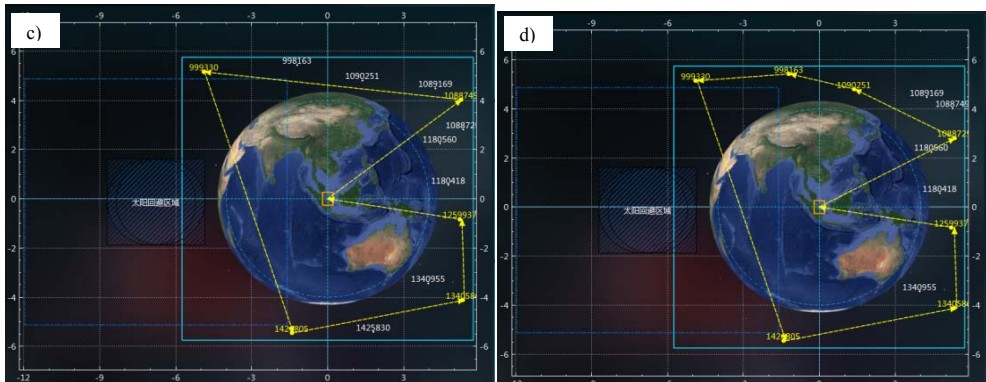

**Fig. 9 GIIRS star selection results - with the sun: (a) 1 star, (b) 3 stars, (c) 5 stars and (d) 7 stars.**

Using the specifically developed software of star observation strategy, statistics of star selection tasks of AGRI and GIIRS

are given in Table 9 and Table 10, respectively. The statistical period is from March, 2017 to March, 2018. Different situations encountered include changing field of view, forecast period, magnitude range, star number, star constellation constraint, and so on. The success rate of software operation is 100%, validating that the proposed strategy can give reasonable star selection results in all kinds of situations. And based on these star observation instructions, AGRI and GIIRS have implemented more than 35000 and 35000 in-orbit star observations, respectively, providing precious data for thermal

elastic deformation parameter calculation.

**Table 9 Statistics of AGRI star selection tasks.**

| No. | Number of stars | Operation success rate | Percentage of all tasks |
|-----|-----------------|------------------------|-------------------------|
| 1 | <3 | 100% | 0.67% |
| 2 | 3 | 100% | 3.05% |
| 3 | 4 | 100% | 1.53% |
| 4 | 5 | 100% | 94.75% |

**Table 10 Statistics of GIIRS star selection tasks.**

| No. | Number of stars | Operation success rate | Percentage of all tasks |
|-----|-----------------|------------------------|-------------------------|
| 1 | <3 | 100% | 0.16% |
| 2 | 3 | 100% | 3.42% |
| 3 | 4 | 100% | 0.76% |
| 4 | 5 | 100% | 95.66% |

As mentioned above, the moon's position in the field of view is predicted precisely and automatically. Moon task will be inserted into the time schedule while the moon is forecasted to be suitable for observation. From March, 2017 to March,

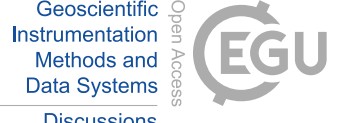



2018, in all 194 AGRI moon tasks and 106 GIIRS moon tasks are arranged automatically through the software of instrument observation strategies. The details are listed in Table 11 and Table 12. Nearly 100% success rate validates the effectiveness of the moon forecasts as well as the instrument observation strategies. The occasional unobserved task was because the visible part of the moon was so small that it was invisible in the images.

**Table 11 Statistics of AGRI moon tasks.**

| Time | Tasks arranged | Observation success rate |
|------|----------------|--------------------------|
| 2017-03 | 12 | 100% |
| 2017-04 | 20 | 95% |
| 2017-05 | 38 | 100% |
| 2017-06 | 16 | 100% |
| 2017-07 | 18 | 100% |
| 2017-08 | 18 | 100% |
| 2017-09 | 7 | 100% |
| 2017-10 | 12 | 100% |
| 2017-11 | 18 | 100% |
| 2017-12 | 19 | 100% |
| 2018-01 | 7 | 100% |
| 2018-02 | 6 | 100% |
| 2018-03 | 3 | 100% |

**Table 12 Statistics of GIIRS moon tasks.**

| Time | Tasks arranged | Observation success rate |
|------|----------------|--------------------------|
| 2017-03 | 8 | 100% |
| 2017-04 | 0 | -- |
| 2017-05 | 2 | 100% |
| 2017-06 | 11 | 100% |
| 2017-07 | 16 | 100% |
| 2017-08 | 16 | 100% |
| 2017-09 | 5 | 100% |
| 2017-10 | 8 | 100% |
| 2017-11 | 7 | 100% |
| 2017-12 | 13 | 100% |
| 2018-01 | 8 | 100% |
| 2018-02 | 6 | 100% |
| 2018-03 | 6 | 100% |

**6 Conclusions**

This paper proposed instrument observation strategies specially designed for AGRI and GIIRS onboard FY-4A satellite, the first satellite of Chinese three-axis stabilized geostationary satellite series. The requirements of navigation, calibration and the earth observation, besides observation flexibility brought by three-axis stabilized platform, are synthetically considered. The most complicated part, star observation strategies, are proposed to select proper stars to be observed by the instruments, whose information is essential to accurate image navigation. Both simulation results and in-orbit application results are given,

including instrument observation strategies, star observation strategies and moon tasks, showing the validity of the proposed



observation strategies. The strategies have been successfully used in FY-4A in-orbit test for more than a year, helping to accomplish more than 163000 and 169000 tasks of AGRI and GIIRS, respectively, laying important foundations for the instruments' daily operation.

So far the star observation of FY-4A has been concentrated on visible stars. With the development of satellites and

5    instruments, star observation of other bands should also be considered in the future. The star observation strategy needs to be specially designed for each observation band according to the band's characteristics. Magnitude limitation, star observation number, angel deviation, etc., should be modified. A whole set of star observation strategies must be proposed for the bands' registration as well as high precision navigation. And instrument observation strategy will also be more complicated for the follow-up instruments.

*Data availability.* The research data can be accessed by direct request to the author.

*Author contributions.* JS, LY and ZZ designed the system. PH, HY, SZ and XC developed the software. CL, JW and LZ took part in the discussion. JS and HY processed the data. JS prepared the manuscript with contributions from all co-authors..

*Competing interests.* The authors declare that they have no conflict of interest.

*Acknowledgments.* We would like to thank Dr. L. Ding, G. Wang and C. Han from Shanghai Institute of Technical Physics, Chinese Academy of Sciences, for their excellent instrument design and helpful discussions. We are grateful to the entire

FY-4 team of National Satellite Meteorological Center, China Meteorological Administration for durable and in-depth research on the ground system of FY-4 satellite. We would also like to thank the reviewers for helping us to improve the paper. This work was supported by the National Natural Science Foundation of China (61172113 and 91338109).

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
