# Peer review of "Instrument observation strategy of new generation three-axis stabilized geostationary meteorological satellite of China"

_Geoscientific Instrumentation, Methods and Data Systems, 2018_

## Referee Comment (RC1) · Anonymous Referee #1 · 26 Apr 2019

General Comments: This manuscript presents simulation results and in-orbit results for three-axis stabilized geostationary satellite attitude control. The geostationary satellite uses multiple stars as a reference to stabilize the platform for acquiring latitude, longitude, and altitude of image. The simulation results of star observational studies are applied to the in-orbit application. The simulation methodology should be described more clearly step-by-step. A detailed description of the simulation procedure will help the reader to appreciate the results presented in tables 3, 4,5 and 6.

The figure captions should be more descriptive. The axis labels in the figures are missing.

[Figure]

**GID**

This study has a practical significance, but the authors did not provide a mathematical framework and numerical example of the simulation. . Specific and General Comments:

In the document the phrase 'so on' should either be deleted or replace it the with necessary description.

Page 1, line #19: Define 'moon tasks.'

Page 1, line#26: Briefly describe the increase in observation efficiency and flexibility compared to FY-2

Page 1, line#27: What are the challenges in image navigation and registration in the case of three-axis stabilized satellite?

Page 1, line#29: Describe edge detection and how it is used in a spin-stabilized satellite.

Page 2, line #1: list other important parameters.

Page 2, line #2: What is the range of variation in space thermal source variation?

Page 2, line #3-4: Rewrite the sentence.

Page 2, line #5: State the expected thermal gradient across the three-axis stabilized the platform.

Page 2, line #5: What is meant by launch violation?

---

## Referee Comment (RC2) · Anonymous Referee #2 · 28 Apr 2019

General Comments: The manuscript mainly focuses on the observation stratedy design of FY-4 satellite and proposed complete and abundant respects towards the onboard imager and sounder. It has significant engineering implementation value, not only for meteorological satellite, but for a variety of remote sensing satellites.

Specific Comments:

1. Please add an introduction of the instruments to help the readers to understand the background, including the bands and resolution.

2. The detector array of star observation should be provided as auxiliary explanation, before the star observation stratedies are proposed.

3. The star observation strategy can be analyzed based on instrument parameters and characteristics, respectively in page 5 and page 7.

4. The overall instrument observation stratedy, including full disk observation, black-body observation, star observation, et al., should be described and summarized in a table or figure clearly in page 8 before the simulation.

---

## Author Comment (AC1) · 8 May 2019

The manuscript has been revised earnestly according to the reviewer's suggestions. Thanks very much for the detailed guidance. The main modifications are listed below, and the revised manuscript is also submitted. Thank you again.

Question 1: The simulation methodology should be described more clearly step-by-step. A detailed description of the simulation procedure will help the reader to appreciate the results presented in tables 3, 4,5 and 6.

Answer: An accurate star forecasting module is specially developed for FY-4A. Using

coordinate system transformation and time system transformation, the star positions in ICRS are transformed to scanning angles and stepping angles in instrument coordinate system. Then judge whether the stars can be observed in the instruments' field of view at the given observation time. Selected and record the suitable stars with its magnitude and position. Based on these forecasting results, many aspects of star observation strategies can be analyzed. An important constrain of star observation strategy is star magnitude. The percentages of different star magnitude ranges of AGRI and GIIRS are counted for every month, which are given in the corresponding tables in the document. The larger the magnitude, the more stars there are. But the smaller the magnitude is, the higher the accuracy of star centroid extraction may achieve. Within the range of 3.5 to 6.5/7.5, which is determined by AGRI/GIIRS detecting ability, a few stars' magnitude lies in the range of 3.5 to 4.5, which are more likely to be selected in star observation strategy. Moreover, distribution and many thresholds mentioned above must be considered comprehensively.

Question 2: The figure captions should be more descriptive. The axis labels in the figures are missing.

Answer: The figure captions are modified to be more descriptive. Some figures are provided by the operational system with a specific format and the axis are described in the text to help the readers to understand the figures better.

Question 3: This study has a practical significance, but the authors did not provide a mathematical framework and numerical example of the simulation.

Answer: This study is a foundation of FY-4 image navigation work concerning task planning, mainly focusing on the instrument observation strategies as well as automatic star observation strategies establishment. The strategies are designed based on analyzing the instruments' characteristics, earth observation requirement, navigation requirement and calibration requirement. And the simulation is carried out based on star forecasting and star observation strategies developed specially for AGRI and

GIIRS. The navigation accuracy of AGRI can be estimated with star observation as follows. And it is similar for GIIRS.

The navigation accuracy is theoretically determined by the accuracy of attitude measurement, orbit determination, scan mirror pointing control, star centroid extraction and thermal elastic deformation parameter calculation. The attitude measurement error provided by star sensors is about 3". The orbit determination error is less than 1". The pointing control error of instrument scan mirror is 3". The star centroid extraction error can be better than 1". The calculation error of thermal elastic deformation parameter is less than 14". Thus the navigation accuracy is about

$$\sqrt{(3^2+1^2+3^2+1^2+14^2)}=14.70$$

14.70" corresponds to 2.54 kilometers on the ground observed from the geostationary orbit. This navigation accuracy obtained by star observation can satisfy the index of 1 infrared pixel, which corresponds to 4 kilometers on the ground.

Question 4: In the document the phrase 'so on' should either be deleted or replace it the with necessary description.

Answer: The phrase 'so on' is thoroughly checked through the document. Some are deleted and others are supplemented with necessary descriptions.

Question 5: Page 1, line #19: Define 'moon tasks.'

Answer: It means moon observation tasks. The moon is forecasted first to offer the accurate time and position in the instrument's field of view, and then the instrument is commanded to automatically observe the moon at the given time and at the given position. That's a complete moon task.

Question 6: Page 1, line#26: Briefly describe the increase in observation efficiency and flexibility compared to FY-2.

Answer: Fengyun-4 (FY-4) is China's new generation geostationary meteorological

satellite series, which is three-axis stabilized instead of spin stabilized as in Fengyun-2 (FY-2) satellites. The three-axis stabilized attitude control mode can effectively increase observation efficiency and flexibility as the satellite can make observations at any time, while the spin stabilized satellite can only observe the earth when it sweeps across it. The full disk observation time is shorted from 30 to 15 minutes. And many different kinds of observation tasks can be designed for FY-4, which is impossible for FY-2.

Question 7: Page 1, line#27: What are the challenges in image navigation and registration in the case of three-axis stabilized satellite?

Answer: The space thermal source changes enormously and the main forms of thermal conduction are radiation and conduction. Lacking of important convection makes the thermal environment of space orbit very abominable. A spin stabilized satellite tends to equalize the thermal variation seen by the instrument over the day, whereas the thermal gradients across the three-axis stabilized platform are more extreme. This can introduce thermal distortions in the platform structure causing changes in the instrument to platform alignment, which will result in navigation error. And this makes the navigation of three-axis stabilized satellite more complicated.

Question 8: Page 1, line#29: Describe edge detection and how it is used in a spin-stabilized satellite.

Answer: Edge detection is the core of landmark navigation, which is often used in the navigation of spin stabilized satellites. Firstly, recognize visible landmarks such as coastline and lakes in the image. Secondly, conduct edge detection using the recognized landmarks and the ideal maps to obtain the landmark matching bias. Thirdly, compute the satellite attitude or instrument viewing vectors using the result of edge detection, that is, the landmark matching bias. But the navigation accuracy of edge detection is limited, and thus is not suitable for the navigation of three-axis stabilized satellite.

Question 9: Page 2, line #1: list other important parameters.

Answer: Other important parameters mainly are auxiliary angle information, which include satellite azimuth angle, satellite zenith angle, solar azimuth angle and solar zenith angle.

Question 10: Page 2, line #2: What is the range of variation in space thermal source variation?

Answer: The thermal deformation ranges from zero to more than a thousand micro arc according to the in-orbit test result of FY-4A satellite. If this huge deformation was not recognized and compensated, it would result in great navigation error.

Question 11: Page 2, line #3-4: Rewrite the sentence.

Answer: The thermal environment of space orbit is very abominable because it lacks important convection.

Question 12: Page 2, line #5: State the expected thermal gradient across the three-axis stabilized the platform.

Answer: The thermal gradient across the three-axis stabilized platform changes over time and it affects the instrument's viewing vector. It can exceed 200 micro arc in an hour according to the in-orbit test result of FY-4A satellite, which can result in nearly 10 kilometers navigation bias in the geostationary orbit.

Question 13: Page 2, line #5: What is meant by launch violation?

Answer: We are very sorry for the spelling mistake. It should be "launch vibration". Launch vibration means the vibration brought by the launch process of the satellite, including low frequency vibration and high frequency vibration. The satellite structure and instruments can be infected by the vibrant condition.

Please also note the supplement to this comment:
https://www.geosci-instrum-method-data-syst-discuss.net/gi-2018-35/gi-2018-35-AC1-supplement.zip

---

## Author Comment (AC2) · 8 May 2019

The manuscript has been revised earnestly according to the reviewer's suggestions. Thanks very much for the detailed guidance. The main modifications are listed below, and the revised manuscript is also submitted. Thank you again.

Question 1: Please add an introduction of the instruments to help the readers to understand the background, including the bands and resolution.

Answer: Geostationary imager is the most effective instrument onboard the geostationary satellites all over the world. AGRI is the core instrument of FY-4A satellite,

[Figure]

which aims to carry out high temporal and spatial resolution imaging of the atmosphere, cloud, land and ocean, in over 14 spectral bands in visible (VIS), near infrared (NIR) and infrared (IR) spectral regions, providing important information for weather analysis and forecast, climate research, environment and disaster monitoring. Main parameters of AGRI are shown is Table 1, and the information of its 14 bands is summarized in Table 2.

Table 1: Main parameters of AGRI. (Please see the supplement.)

Table 2: Spectral configuration of AGRI. (Please see the supplement.)

GIIRS is another important instrument onboard FY-4A. Its main objective is to detect atmospheric temperature, moisture and trace gas content precisely, providing input data for numerical weather forecast, disastrous weather monitoring and atmospheric chemical composition detection. It has visible, medium / shortwave and longwave IR bands. Main parameters of GIIRS are shown in Table 4 of the manuscript.

Table 4: Main parameters of GIIRS. (Please see the supplement.)

Question 2: The detector array of star observation should be provided as auxiliary explanation, before the star observation strategies are proposed.

Answer: AGRI has 14 observation bands, the second (0.55-0.75$\mu$m) of which is designed to sense stars of magnitude higher than 6.0, as shown in Table 2. The detector size is 32 (north-south direction) by 4 (west-east direction), with a gap between adjacent columns, which is shown is Fig. 1. This must be considered in developing AGRI's star observation strategy to guide its in-orbit daily star sensing.

Figure 1: Detector array diagram of AGRI. (Please see the supplement.)

GIIRS is designed deliberately with a visible band (0.55-0.75$\mu$m) to sense stars of magnitude lower than 6.5. The detector size is 330 (north-south direction) by 256 (west-east direction) and no gap between adjacent columns as well as rows, which is shown is Fig. 3 of the manuscript. Star observation strategy of GIIRS must also be developed, considering both the similarities and differences between AGRI and GIIRS. The similarities include requirements of star centroid extraction as well as thermal elastic deformation parameter calculation. The main difference comes from the different instantaneous field of view (IFOV), which will be analyzed in detail.

Figure 3: Detector array diagram of GIIRS. (Please see the supplement.)

Question 3: The star observation strategy can be analyzed based on instrument parameters and characteristics, respectively in page 5 and page 7.

Answer: For AGRI, the detector size is 32 by 4, with a gap between adjacent columns, which is shown in Fig. 1. This must be considered in developing AGRI's star observation strategy. Dwell observation mode is recommended for star sensing, waiting for the star crossing the whole focal plane, aiming at obtaining star observation data of relative long time series. This can effectively improve the accuracy of star centroid extraction. AGRI's angular resolution determines the minimal distance between any two candidate stars. Another characteristic of AGRI is its small detector size, which means most of the time only one star can be observed. This directly influences the minimal distance between the candidate star and another star.

For GIIRS, its star sensing ability is higher than AGRI, which means there are more stars that can be observed. And its detector size is larger, as shown in Fig. 3, which means not only one star but also multiple stars can be observed at a time. Thus multiple star selection strategy should first be considered because this is advantageous for star recognition. Another characteristic of GIIRS is it can't wait for the star crossing the whole focal plane, as its detector is so large. A small area of less than 10 pixels near the center of the detector is chosen to be used for star sensing, in order to save observation time as well as to avoid the effect of distortion at the edge of the detector.

Thus two specific star observation strategies are developed respectively for the two instruments, based on their similarities and differences.

Question 4: The overall instrument observation strategy, including full disk observation, blackbody observation, star observation, et al., should be described and summarized in a table or figure clearly in page 8 before the simulation.

Answer: The AGRI observation strategy and GIIRS observation strategy are proposed above in detail. In every time block, one full disk task or three China region tasks, one infrared background task, one blackbody task and one star task are arranged for AGRI, while one region task, one blackbody task, one cold space task and one star task are arranged for GIIRS. The time block is shown if Fig. 4 for clarity. FY-4A satellite's daily operation is implemented based on these two time blocks.

Figure 4: Time blocks for AGRI and GIIRS used in daily operation: (a) AGRI and (b) GIIRS. (Please see the supplement.)

Please also note the supplement to this comment:
https://www.geosci-instrum-method-data-syst-discuss.net/gi-2018-35/gi-2018-35-AC2-supplement.zip
* * *
[Figure]

Figure 1: Detector array diagram of AGRI.

**Fig. 1.** Detector array diagram of AGRI

X

1   2   3   ......   256

...
...

329

330

Y

Figure 3: Detector array diagram of GIIRS.

**Fig. 2.** Detector array diagram of GIIRS

a)

| Full disk observation / China region observation | → | Infrared background observation | → | Blackbody observation | → | Star observation |

b)

| Region observation | → | Cold space observation | → | Blackbody observation | → | Star observation |

**Figure 4: Time blocks for AGRI and GIIRS used in daily operation: (a) AGRI and (b) GIIRS.**

**Fig. 3.** Time blocks for AGRI and GIIRS used in daily operation